# Changes of bioactive components and antioxidant potential during fruit development of *Prunus humilis* Bunge

**Hongbo Fu**[1,2], **Yujia Qiao**[3], **Pengfei Wang**[1], **Xiaopeng Mu**[1], **Jiancheng Zhang**[1], **Baochun Fu**[2], **Junjie Du**[1] *

1 College of Horticulture, Shanxi Agricultural University, Taigu, Shanxi, People's Republic of China,
2 Research Institute of Pomology, Shanxi Agricultural University, Taigu, Shanxi, People's Republic of China,
3 Research Institute of Wheat, Shanxi Agricultural University, Linfen, Shanxi, People's Republic of China

* djj738@163.com

**Data Availability Statement:** All relevant data are within the paper and its Supporting Information files.

## Abstract

Dynamic changes in flavonoid, total phenol, and antioxidant potential in different *Prunus humilis* accessions during fruit development stages were studied in order to provide a reference for the optimum harvest time for flavonoid extraction. 'Nongda 4', 'Nongda 5', 'DS-1' and '02–16' were selected as plant materials to determine the content of flavonoid, total phenol and antioxidant indices during six fruit development stages. Changes in total flavonoid content (TFC) and total phenol content (TPC) in different accessions of *P. humilis* were slightly different depending on the development stage of *P. humilis* fruit. TFC and TPC in 'Nongda 5' fruit showed a trend of continuous decline. There was a small increase in TFC and TPC from the young fruit stage to the stone hardening stage, followed by a decreasing trend, and then to the lowest level at the ripening stage of 'Nongda 4', 'DS-1', and '02–16' fruits. The trend of antioxidant capacity (ABTS, FRAP, DPPH) with the TFC and TPC of *P. humilis* fruit was basically the same, and the correlation analysis results showed that the TFC of *P. humilis* fruit was positively correlated with the antioxidant indices (*P*<0.01). Catechin (CC), rutin (RT), and quercetin-7-O-β-D-glucopyranoside (Q7G) were detected in all the fruit development stages of the four *P. humilis* fruits. Among them, catechin was the most abundant component, accounting for approximately 10%. Myricetin (MC) and quercetin (QC) were generally detected only in the early fruit development stage, but not in the later fruit development stage. Correlation analysis showed that the flavonoid components with TFC, TPC, and antioxidant indices differed between the different accessions. RT, CC, and liquiritigenin (LR) had a stronger correlation with TFC and antioxidant indices. Cyanidin-3-O-glucoside (C3G) was not detected until the coloring stage in two red *P. humilis* accessions ('Nongda 4' and 'DS-1'), and so it is better to choose a red *P. humilis* fruit to extract C3G at the ripening stage. Selecting an early stage of fruit development, especially the stone hardening stage, was important for extracting flavonoids, total phenols and other components. We believe that our results will provide basic information and reference for evaluation of fruit nutrition and health benefits, breeding of functional new varieties, and efficient utilization of *P. humilis* fruit.

**Funding:** This work was supported by: The 191 province 1331 jinou 1 (Grant No.J201911313), the Key Projects of Key Research and Development of Shanxi Province (Grant No. 201703D211001-04-04), and the Applied Basic Research Project of Shanxi Province (Grant No. 201801D121251).

**Competing interests:** The authors have declared that no competing interests exist.

## Introduction

*Prunus humilis* Bunge (*Cerasus humilis* (Bunge) S.Ya.Sokolov) (Rosaceae) is a small deciduous shrub [1]. It is an ancient tree species in China for approximately 3000 years [2]. The fruits are rich in vitamins, organic acids, and mineral elements and contain relatively high levels of anthocyanins, flavonoids, phenols, and other bioactive components [3]. In China the fruits are named 'calcium fruit' because of their high calcium content [4]. This plant has both ecological and economic benefits. Flavonoids (e.g., flavones, isoflavones, flavanols, and anthocyanins) are important secondary metabolites in plants [5]. Flavonoids benefit the human body due to their antioxidant and antiaging properties. Although the total flavonoid content, polyphenol compounds, and radical scavenging activity of *P. humilis* in Shanxi province [6, 7], Liaoning province [8], Beijing city [3] and other places [9] have been evaluated, fruit was evaluated only at the ripening stage of development. To the best of our knowledge, this is the first study to evaluate the dynamic changes in bioactive components and antioxidant potential of *P. humilis* accessions. We selected four *P. humilis* accessions during six fruit development stages and analyzed the total flavonoid content, total phenol content, bioactive compounds, and antioxidant capacities of the fruit in order to understand the dynamic change regulation of flavonoids in *P. humilis* fruit, information that could be beneficial for the development of functional products with different nutritional and health benefits. The results may also provide a guarantee for further studies on the mechanism of flavonoid synthesis, accumulation, and distribution in *P. humilis* fruit.

## Materials and methods

### Plant materials

Four *P. humilis* accessions ('Nongda 4', 'Nongda 5', '02–16', and 'DS-1') were selected as experimental materials and acquired from the germplasm nursery of Shanxi Agricultural University, Jinzhong, China (37°23′N, 112°29′E). Six fruit development stages (young fruit stage, stone hardening stage, expension stage, turning stage, coloring stage and ripening stage) were acquired for per accession in 2019 (Table 1). A total of three biological replicated consisting in 3 healthy plants each one was selected. Fifteen pest- and disease-free fruits were sampled from the top, middle, and bottom of each sampled plants and stored at -40°C before analysis.

### Chemicals

Methanol and acetonitrile used in ultrahigh performance liquid chromatography (UHPLC) were purchased from OmniGene LLC (Morrisville, NC, USA). Analytical flavonoid standards, i.e., catechin, myricetin, epicatechin, rutin, quercetin, liquiritigenin, cyanidin-3-O-glucoside, gallic acid, Trolox, 2,2'-azino-bis(3-ethylbenzothiazoline-6-sulfonic acid (ABTS),

**Table 1. Sampling date of *Prunus humilis* during fruit development stage in 2019.**

|  | 'Nongda 4' | 'Nongda 5' | '02–16' | 'DS-1' |
| --- | --- | --- | --- | --- |
| Young fruit stage (YF) | May 25 | May 25 | May 25 | May 25 |
| Stone hardening stage (SH) | Jun 22 | Jun 22 | Jun 15 | Jun 5 |
| Expension stage (ES) | Jul 20 | Jul 6 | Jun 30 | Jun 25 |
| Turning stage (TS) | Aug 1 | Jul 20 | Jul 15 | Jul 5 |
| Coloring stage (CS) | Aug 15 | Aug 1 | Jul 30 | Jul 12 |
| Ripening stage (RS) | Aug 30 | Aug 15 | Aug 13 | Jul 19 |

1,1-diphenyl-2-picrylhydrazyl (DPPH), quercetin-7-O-β-D-glucopyranoside, and 2,4,6-tri-(2-pyridyl)-1,3,5-triazine (TPTZ) were purchased from Solarbio Technology Co. Ltd. (Beijing, China).

## Total flavonoid and phenol content

The extracts were prepared according to the methods of Fu et al. [7]. All the sampled fruits were pulverized in liquid nitrogen and extracted with 40% (v/v) acidified methanol. The ratio of material to solution was 1:10. The suspension was mixed by whirlpool oscillation, extracted by ultrasound at 40 kHz for 30 min, and centrifuged at $10\,000 \times g$ for 15 min at 4˚C. The extraction was repeated thrice, and the filtrates were pooled.

Total flavonoid content (TFC) was determined by $NaNO_2$-$Al(NO_3)_3$ colorimetry [7]. First, 0.8 mL of the extracted solution was transferred to a 10 mL volumetric flask containing 0.3 mL of 5% (w/v) $NaNO_2$. The mixture was shaken well and kept in the dark for 6 min. Thereafter, 0.3 mL of 10% (w/v) $Al(NO_3)_3$ was added to the mixture, shaken well, and kept in the dark for another 6 min. Then, 4 mL of 4% (w/v) NaOH was added to the mixture and shaken well. The volume was adjusted to 10 mL using 40% (v/v) methanol, and the mixture was shaken well and kept in the dark for 10 min. Finally, the absorbance was read at 510 nm in a spectrophotometer (UV-5200 Shanghai Yuanxi Instrument Co. Ltd., Shanghai, China) calibrated with a rutin standard curve ranged from 0 mg/ml to 0.1 mg/ml.

Total phenol content (TPC) was determined by Folin-Ciocalteu colorimetry. First, 0.2 mL of the extracted solution was added to a 10 mL volumetric flask containing 0.2 mL Folin-Ciocalteu reagent. The mixture was shaken well and left to stand for 4 min. Then, 1 mL of 10% (w/v) $Na_2CO_3$ was added to the mixture, and the volume was adjusted to 8 mL with $ddH_2O$. The mixture was shaken well and incubated in a water bath at 35˚C for 1 h. The absorbance was read at 760 nm in a spectrophotometer (UV-5200 Shanghai Yuanxi Instrument Co. Ltd., Shanghai, China) calibrated using a gallic acid standard curve.

## Antioxidant capacity

**DPPH assay.** Following the method of Fu et al. [7], we added 2.8 mL of 0.1 mM DPPH to 0.2 mL of extracted solution. The mixture was shaken well and kept in the dark for 30 min at 25˚C. For the blank, the extracted solution was replaced with 40% (v/v) methanol. The absorbance was read at 517 nm in a spectrophotometer (UV-5200 Shanghai Yuanxi Instrument Co. Ltd., Shanghai, China) calibrated using a trolox standard curve.

**Ferric reducing antioxidant power (FRAP) assay.** Following the method of Fu et al. [7], we added 4.9 mL of FRAP solution (0.1 M $CH_3COONa$ [pH = 3.6], 10 mM TPTZ, and 20 mM $FeCl_3$ in a 10:1:1 volumetric ratio) to 0.1 mL of extracted solution. The sample mixture was shaken well and incubated in the dark for 10 min at room temperature. For the blank, the extracted solution was replaced with 40% (v/v) methanol. The absorbance was read at 593 nm in a spectrophotometer (UV-5200 Shanghai Yuanxi Instrument Co. Ltd., Shanghai, China) calibrated using a trolox standard curve.

**ABTS assay.** Following the method of Fu et al. [7], we added 3.9 mL of $ABTS^+$ solution (7 mL ABTS plus 140 mM $K_2(SO_4)$) to 0.1 mL of extracted solution. The $ABTS^+$ solution was prepared in the dark at 25˚C over 12–16 h. The sample mixture was shaken well and incubated in the dark for 10 min at room temperature. For the blank, the extracted solution was replaced with 40% (v/v) methanol. The absorbance was read at 734 nm in a spectrophotometer (UV-5200 Shanghai Yuanxi Instrument Co. Ltd., Shanghai, China) calibrated using a trolox standard curve.

### Flavonoid components

The content of eight flavonoid components was determined by UHPLC in an Agela Venusil ABS C18 column (4.6 mm × 250 mm; 5 μm) (Agela Technologies, Wilmington, DE, USA). In particular, 1 mL of flavonoid extract was passed through a 0.22 μm Millipore membrane filter (EMD Millipore, Burlington, MA, USA) and placed in a liquid sample bottle. The solvent system consisted of 0.5% (v/v) formic acid water (solvent A) and acetonitrile (solvent B). The flow rate was set to 0.8 mL min$^{-1}$, and the run time was 69 min. The sample injection volume was 20 μL. The gradient program was as follows: 90% A at 0 min; 87% A for 0–5 min; 84% A for 5–25 min; 79% A for 25–30 min; 78% A for 30–45 min; 76% A for 45–50 min; 75% A for 50–65 min; and 90% A for 65–69 min. The detector was set to 280 nm (for detecting catechin, epicatechin, and liquiritigenin), 360 nm (rutin, myricetin, quercetin and quercetin-7-O-β-D-glucopyranoside), and 520 nm (cyanidin-3-O-glucoside) for the simultaneous monitoring of the various flavonoid components [7].

### Statistical analysis

Data were analyzed in Microsoft Excel v. 2007 (Microsoft Corporation, Redmond, WA, USA). Correlation analysis, principal component analysis and differences among the mean values were evaluated using ANOM (analysis of means), and statistical significance was set at $P < 0.05$ using the Statistical Analysis System v. 9.2 (SAS Institute, Cary, NC, USA). Tbtools (http://doi.org/10.1101/289660) and OriginPro9.0 (OriginLab Corporation, Northampton, MA, USA) were used to draw figures. Total flavonoid content (TFC) = $C_1 \times (10/V_1) \times V_2 \times m^{-1}$, total phenol content (TPC) = $C_2 \times (8/V_1) \times V_2 \times m^{-1}$, antioxidant indices (DPPH, FRAP, and ABTS) = $C_3 \times V_1 \times V_2 \times m^{-1} \times 1000$. $C_1$ indicates the concentration of rutin, $C_2$ of gallic acid, $C_3$ of trolox, $V_1$ the volume of reaction solution, $V_2$ the volume of extraction solution, and m the fruit weight.

## Results

### Dynamic change of total flavonoid content in *Prunus humilis* fruit

The flavonoid content of *P. humilis* fruit showed a decrease in the fruit development stage, but there were differences among the different accessions (Fig 1). The flavonoid content in 'Nongda 5' fruit decreased continuously from the young fruit stage (56.45 mg/g FW, fresh weight) to the ripening stage (6.49 mg/g FW) ($P < 0.05$). Compared with the other three accessions ('Nongda 4', '02–16', and 'DS-1'), 'Nongda 5' had a low flavonoid content level at all fruit development stages. The flavonoid content of 'Nongda 4', '02–16', and 'DS-1' increased slightly from the young fruit stage to the stone hardening stage, which had the highest flavonoid value of any development stage, with differences between accessions ('02–16' was the highest, with 81.86 mg/g FW). The flavonoid content decreased continuously from the stone hardening stage to the ripening stage, and at the ripening stage, '02–16' still had a higher value than the other accessions. Indeed, '02–16' showed a high flavonoid content at all fruit development stages.

### Dynamic change of total phenol content in *Prunus humilis* fruit

The changes in total phenol content in *P. humilis* fruit were consistent with those of total flavonoid content (Fig 2). The phenol content in 'Nongda 5' fruit decreased continuously from the young fruit stage (15.88 mg/g FW) to the ripening stage (2.25 mg/g FW) ($P < 0.05$). Compared with the other three accessions ('Nongda 4', '02–16', and 'DS-1'), 'Nongda 5' had a low phenol content level from the expension stage to the ripening stage. The phenol content of 'Nongda

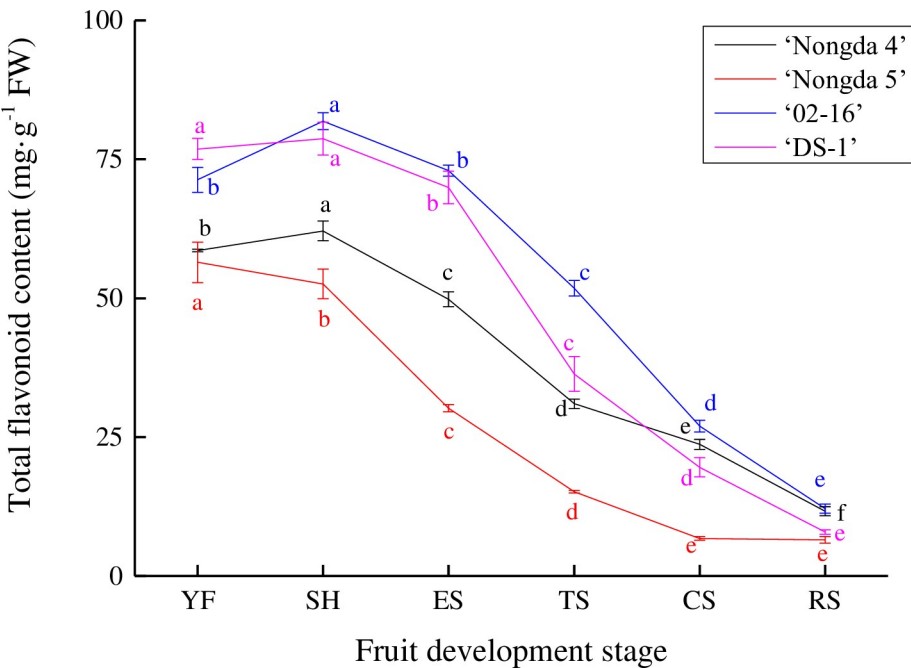

**Fig 1. Dynamic change of total flavonoid content in *Prunus humilis* fruit.** Note: The same color letters indicate significance at $P<0.05$. FW, Fresh weight; YF, Young fruit stage; SH, Stone hardening stage; ES, Expension stage; TS, Turning stage; CS, Coloring stage; RS, Ripening stage.

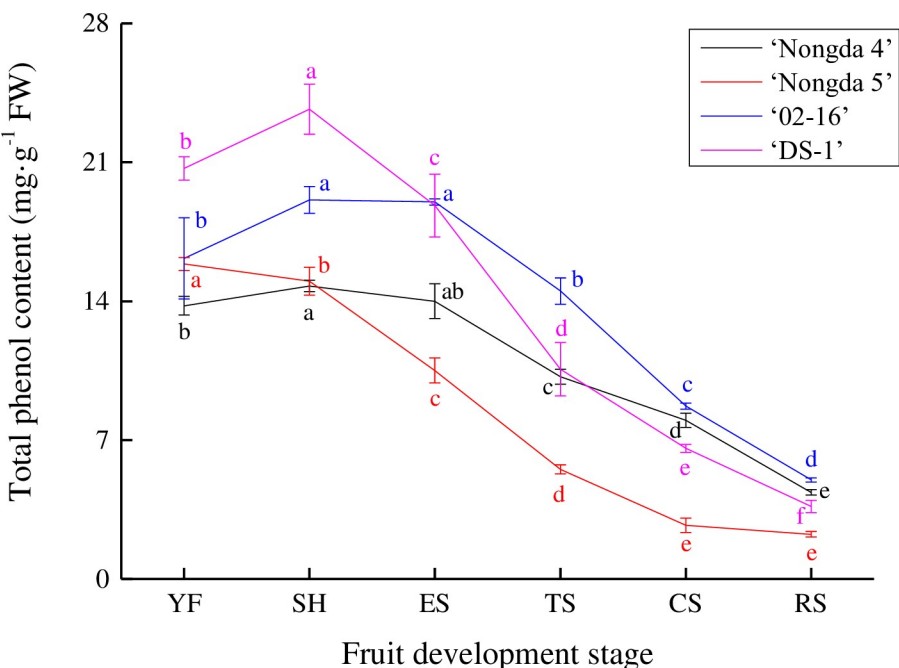

**Fig 2. Dynamic change of total phenol content in *Prunus humilis* fruits.** Note: The same color letters indicate significance at $P<0.05$. FW, Fresh weight; YF, Young fruit stage; SH, Stone hardening stage; ES, Expension stage; TS, Turning stage; CS, Coloring stage; RS, Ripening stage.

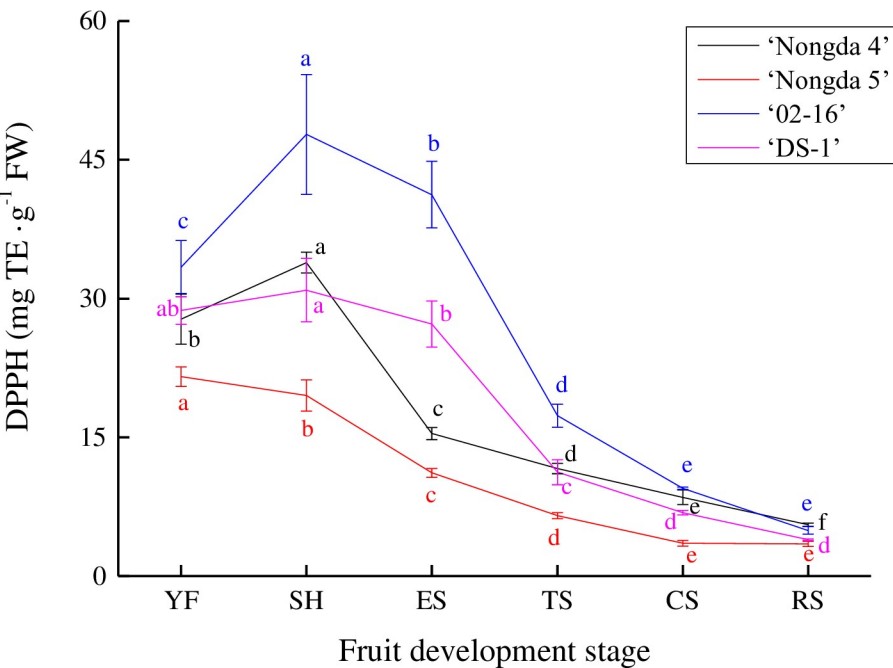

**Fig 3. Dynamic change of DPPH scavenging ability in *Prunus humilis* fruit.** Note: The same color letters indicate significance at *P*<0.05. FW, Fresh weight; TE, Trolox equivalent; DPPH, 2,2-diphenyl-1-picrylhydrazyl free radical scavenging capacity; YF, Young fruit stage; SH, Stone hardening stage; ES, Expension stage; TS, Turning stage; CS, Coloring stage; RS, Ripening stage.

4', '02–16', and 'DS-1' increased slightly from the young fruit stage to the stone hardening stage, which had the highest level of any development stage. With the development of fruit, the total phenol content of 'DS-1' decreased greatly from the stone hardening stage (23.68 mg/g FW) to the ripening stage (3.65 mg/g FW), and decreased by 548.77%. '02–16' and 'Nongda 4' decreased by 282.97% and 238.76% respectively, which was one reason that the total phenol content in '02–16' and 'Nongda 4' fruits was higher than in 'DS-1' at the ripening stage.

### Dynamic change of antioxidant capacity in *Prunus humilis* fruit

The changes in DPPH scavenging ability in 'Nongda 4', 'Nongda 5', '02–16', and 'DS-1' fruits were consistent with the content of flavonoid and total phenol (Fig 3). The DPPH of 'Nongda 5' fruit continued to decrease during all fruit development stages. In 'Nongda 4', '02–16', and 'DS-1' fruits, DPPH increased slightly from the young fruit stage to the stone hardening stage, and then decreased gradually to the ripening stage. All four accessions decreased slowly during the turning stage. The DPPH scavenging ability of the four accessions showed obvious differences from the stone hardening stage to the expension stage, but differences from the coloring stage to the ripening stage were relatively small. At all fruit development stages, 'Nongda 5' had a low DPPH scavenging ability level, and '02–16' had a strong DPPH scavenging ability level.

The ferric reducing antioxidant power (FRAP) of 'Nongda 4', 'Nongda 5', '02–16' and 'DS-1' fruits were 9.89–54.49 mg TE/g FW, 5.41–52.98 mg TE/g FW, 9.79–49.05 mg TE/g FW and 6.88–35.95 mg TE/g FW, respectively (Fig 4). The FRAP of 'Nongda 5' fruit continued to decrease during all fruit development stages and in 'Nongda 4', '02–16', and 'DS-1' fruits, increased slightly from the young fruit stage to the stone hardening stage, then decreased gradually to the ripening stage with a relatively uniform decline. The FRAP between the four

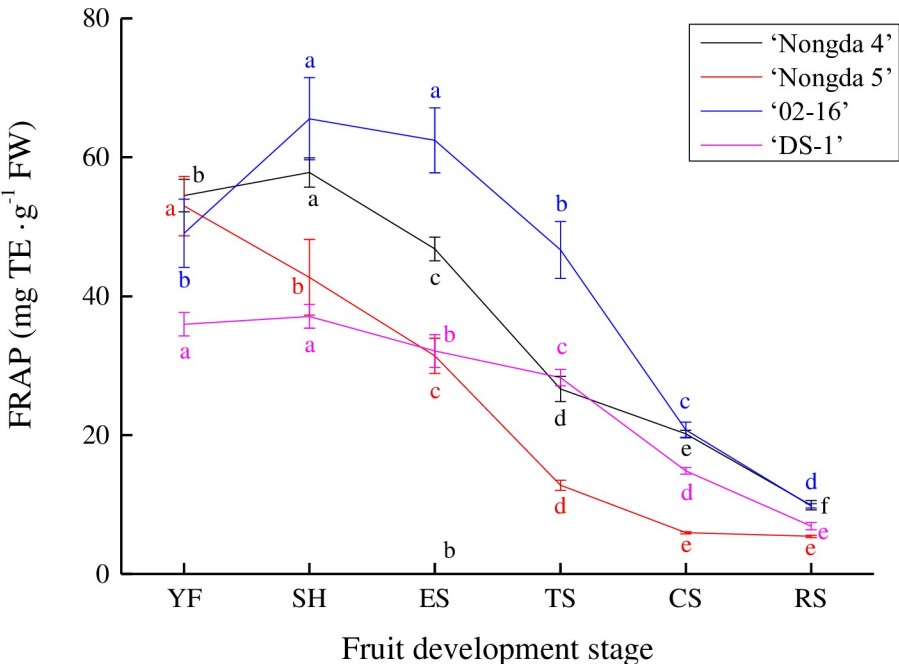

**Fig 4. Dynamic change of ferric reducing antioxidant power in *Prunus humilis* fruit.** Note: The same color letters indicate significance at $P<0.05$. FW, Fresh weight; TE, Trolox equivalent; FRAP, ferric reducing antioxidant power; YF, Young fruit stage; SH, Stone hardening stage; ES, Expension stage; TS, Turning stage; CS, Coloring stage; RS, Ripening stage.

accessions showed obvious differences from the young fruit stage to the coloring stage, and then showed little difference until the ripening stage.

The ABTS scavenging ability of 'Nongda 4', 'Nongda 5', '02–16', and 'DS-1' fruits were 15.09–141.21 mg TE/g FW, 8.20–122.62 mg TE/g FW, 13.13–149.94 mg TE/g FW, and 9.84–117.94 mg TE/g FW, respectively (Fig 5). The ABTS of 'Nongda 5' fruit continued to decrease during all fruit development stages, and in 'Nongda 4', '02–16', and 'DS-1' fruits, increased slightly from the young fruit stage to the stone hardening stage, then decreased gradually to the ripening stage. These fruits showed obvious differences from the young fruit stage to the turning stage, but little difference at the ripening stage.

Taken together, changes in the antioxidant indices (DPPH, FRAP and ABTS) of 'Nongda 4', 'Nongda 5', '02–16', and 'DS-1' fruits were consistent with changes in flavonoid and total phenol content, indicating that the flavonoid and total phenol content of *P. humilis* fruit had strong antioxidant capacity. There were also some differences among the three antioxidant indices, among which ABTS scavenging ability was strongest at all the fruit development stages, followed by FRAP and DPPH, which may be related to the specific flavonoid components in *P. humilis* fruit.

## Dynamic change of flavonoid components in *Prunus humilis* fruits

In 'Nongda 4' fruit, catechin (CC), epicatechin (EC), liquiritigenin (LR), rutin (RT) and quercetin-7-O-β-D-glucopyranoside (Q7G) were detected during all fruit development stages. The highest CC content was 415.71 mg/100 g at the stone hardening stage, and the lowest was 52.26 mg/100 g at the ripening stage, which was significantly different from the other four stages ($P<0.05$) and showed a trend of first increase then gradual decrease (Table 2). The highest EC content was 30.51 mg/100 g at the stone hardening stage which was significantly

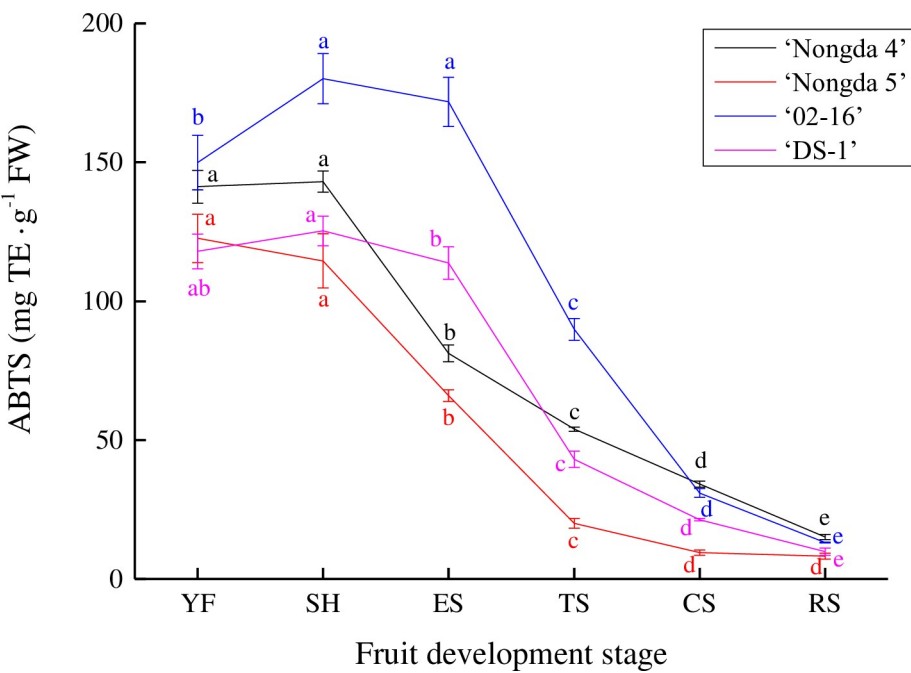

**Fig 5. Dynamic change of ABTS scavenging ability in *Prunus humilis* fruit.** Note: The same color letters indicate significance at *P*<0.05. FW, Fresh weight; TE, Trolox equivalent; ABTS, 2,2'-azinobis (3-ethylbenzothiazoline-6-sulfonic acid) free radical scavenging capacity; YF, Young fruit stage; SH, Stone hardening stage; ES, Expension stage; TS, Turning stage; CS, Coloring stage; RS, Ripening stage.

different (*P*<0.05) from the lowest at the ripening stage, 7.46 mg/100 g. The changing trends of CC and EC were relatively consistent with the total flavonoid content (TFC). The content of Q7G first increased and then decreased and remained steady; at the coloring and ripening stages, Q7G was the lowest (2.93 mg/100 g), significantly different (*P*<0.05) from the highest content (3.81 mg/100 g) at the expension stage. The highest content of LR (4.03 mg/100 g) and RT (4.35 mg/100 g) were both at the young fruit stage and decreased to the ripening stage. Myricetin (MC) and quercetin (QC) were detected only in the young fruit and stone hardening stages. Cyanidin-3-O-glucoside (C3G) accumulated gradually when the fruit turned red from coloring stage (17.77 mg/100 g) to ripening stage (43.50 mg/100 g).

CC, LR, RT and Q7G were detected at all fruit development stages of 'DS-1', while MC and QC were not detected. CC increased gradually from the young fruit stage to the expension stage, but with no significant difference among the first three stages, then decreased to the ripening stage (21.43 mg/100 g) (Table 2). The content of LR and RT were 7.11 mg/100 g and 3.96 mg/100 g respectively at the stone hardening stage, and 1.37 mg/100 g and 1.84 mg/100 g respectively at the ripening stage (*P*<0.05). The changing trend first increased and then gradually decreased, consistent with TFC. Q7G content showed little change during all stages, ranging from 2.39–2.81 mg/100 g. The highest EC content was 27.58 mg/10 0g at stone hardening stage, the lowest content was 4.49 mg/100 g at the coloring stage (*P*<0.05), and EC was not detected at the ripening stage. The regularity of C3G was consistent with that of 'Nongda 4', but the content (18.32 mg/100 g) was lower than 'Nongda 4' fruit at the ripening stage. This may be one of the factors that caused the peel of 'DS-1' to be lighter red than that of 'Nongda 4'.

In '02–16' fruits, CC, EC, LR, RT and Q7G were detected during all fruit development stages, and all five components showed a trend of first increasing and then gradually decreasing. The highest content of RT was 7.64 mg/100 g at the stone hardening stage, and the lowest

**Table 2. Dynamic change of flavonoid components in *Prunus humilis* fruits.**

| Accessions | Fruit development stage | CC (mg/100 g FW) | EC (mg/100 g FW) | LR (mg/100 g FW) | RT (mg/100 g FW) | Q7G (mg/100 g FW) | MC (mg/100 g FW) | QC (mg/100 g FW) | C3G (mg/100 g FW) |
|---|---|---|---|---|---|---|---|---|---|
| 'Nongda 4' | YF | 230.42±16.99[c] | 13.99±0.32[ab] | 4.03±0.16[a] | 4.35±0.07[a] | 3.33±0.02[c] | 1.32±0.01[a] | 7.98±0.17[a] | - |
| | SH | 415.71±30.07[a] | 30.51±10.68[a] | 3.12±0.37[b] | 3.76±0.16[b] | 3.61±0.10[b] | 1.19±0.01[b] | 6.84±0.19[b] | - |
| | ES | 367.98±11.39[b] | 27.92±3.98[a] | 2.17±0.08[c] | 3.32±0.01[c] | 3.81±0.01[a] | - | - | - |
| | TS | 207.94±10.60[c] | 14.03±2.72[ab] | 1.50±0.04[d] | 2.52±0.06[d] | 3.45±0.06[c] | - | - | - |
| | CS | 137.87±13.80[d] | 9.21±4.18[b] | 1.39±0.04[d] | 2.05±0.03[e] | 2.93±0.05[d] | - | - | 17.77±0.47[b] |
| | RS | 52.26±2.42[e] | 7.46±0.78[b] | 1.28±0.03[d] | 1.92±0.06[f] | 2.93±0.13[d] | - | - | 43.50±2.50[a] |
| 'DS-1' | YF | 245.81±25.35[a] | 18.53±10.12[ab] | 6.95±0.37[a] | 3.52±0.40[a] | 2.59±0.17[ab] | - | - | - |
| | SH | 249.40±20.40[a] | 27.58±9.85[a] | 7.11±0.75[a] | 3.96±0.70[a] | 2.62±0.25[ab] | - | - | - |
| | ES | 254.11±21.42[a] | 23.15±9.45[a] | 6.21±0.47[b] | 3.74±0.64[a] | 2.74±0.16[a] | - | - | - |
| | TS | 154.55±23.52[b] | 18.44±3.12[ab] | 4.14±0.32[c] | 3.14±0.67[ab] | 2.81±0.14[a] | - | - | - |
| | CS | 70.42±7.52[c] | 4.49±1.38[b] | 2.34±0.04[d] | 2.52±0.05[b] | 2.53±0.01[ab] | - | - | 13.73±0.01[b] |
| | RS | 21.43±0.48[d] | - | 1.37±0.08[e] | 1.84±0.01[bc] | 2.39±0.06[b] | - | - | 18.32±0.02[a] |
| '02–16' | YF | 383.75±89.87[b] | 21.13±1.74[c] | 7.36±0.15[b] | 5.04±0.42[c] | 2.93±0.05[b] | 1.25±0.05[a] | 5.16±0.15[a] | - |
| | SH | 613.70±56.61[a] | 35.59±1.66[b] | 8.73±0.27[a] | 7.64±0.87[a] | 3.85±0.10[a] | 1.37±0.12[a] | 5.08±0.11[a] | - |
| | ES | 680.84±18.42[a] | 56.00±11.64[a] | 8.93±0.82[a] | 6.62±0.95[b] | 3.98±0.41[a] | 1.25±0.12[a] | 5.06±0.06[a] | - |
| | TS | 397.05±8.84[b] | 18.35±3.93[cd] | 3.34±0.41[c] | 4.31±0.21[c] | 3.76±0.13[a] | 1.10±0.01[a] | - | - |
| | CS | 159.73±9.00[c] | 10.82±1.69[d] | 1.77±0.03[d] | 2.36±0.04[d] | 2.85±0.02[b] | - | - | - |
| | RS | 39.64±2.15[d] | 2.96±0.28[e] | 1.25±0.02[d] | 1.91±0.03[d] | 2.94±0.15[b] | - | - | - |
| 'Nongda 5' | YF | 233.45±6.56[b] | 13.26±0.66[a] | 4.04±0.09[b] | 3.90±0.07[b] | 3.14±0.03[c] | 1.55±0.03[a] | 6.38±0.06 | - |
| | SH | 427.95±9.14[a] | 14.96±0.72[a] | 4.86±0.74[a] | 5.21±0.15[a] | 3.93±0.09[a] | 1.08±0.04[b] | - | - |
| | ES | 207.27±7.28[c] | 15.81±4.36[a] | 2.20±0.03[c] | 2.83±0.03[c] | 3.03±0.02[c] | - | - | - |
| | TS | 125.02±11.30[d] | 7.14±1.24[b] | 1.83±0.06[c] | 2.69±0.09[c] | 3.35±0.09[b] | - | - | - |
| | CS | 41.34±2.49[e] | 1.93±0.07[c] | 1.12±0.01[d] | 1.74±0.02[d] | 2.67±0.03[d] | - | - | - |
| | RS | 28.51±1.44[f] | - | - | 2.07±0.04[d] | 2.59±0.15[d] | - | - | - |

Note: The different letters indicate significance at *P*<0.05 among fruit development stage. Data are means ± SD. CC, catechin; EC, epicatechin; LR, liquiritigenin; RT, rutin; Q7G, quercetin-7-O-β-D-glucopyranoside; MC, myricetin; QC, quercetin; C3G, cyanidin-3-O-glucoside; YF, Young fruit stage; SH, Stone hardening stage; ES, Expension stage; TS, Turning stage; CS, Coloring stage; RS, Ripening stage.

was 1.91 mg/100 g at the ripening stage (*P*<0.05). The content of the other four components was the highest at the expension stage and lowest at the ripening stage (Table 2). MC was detected in the first four stages, and the content showed a decreasing trend, but the difference was not significant. QC content also decreased gradually in the first three stages, but with no significant difference. C3G was not detected during any developmental stage.

In 'Nongda 5' fruit, only three components (CC, RT and Q7G) were detected in every development stage. CC increased gradually from the young fruit stage to the expension stage (427.95 mg/100 g), then decreased to the ripening stage (28.51 mg/100 g), and there were significant differences among all fruit development stages (*P*<0.05) (Table 2). EC and LR first increased and then decreased and were not detected at the ripening stage. MC was detected in the young fruit and stone hardening stages. QC was detected only at the young fruit stage. C3G was not detected during the entire development.

## Correlation analysis of bioactive components and antioxidant indices in different *Prunus humilis* accessions during fruit development stages

The total flavonoid content (TFC) of the four *P. humilis* fruits was significantly positively correlated with the three antioxidant indices (*P*<0.01), but the correlation coefficients were

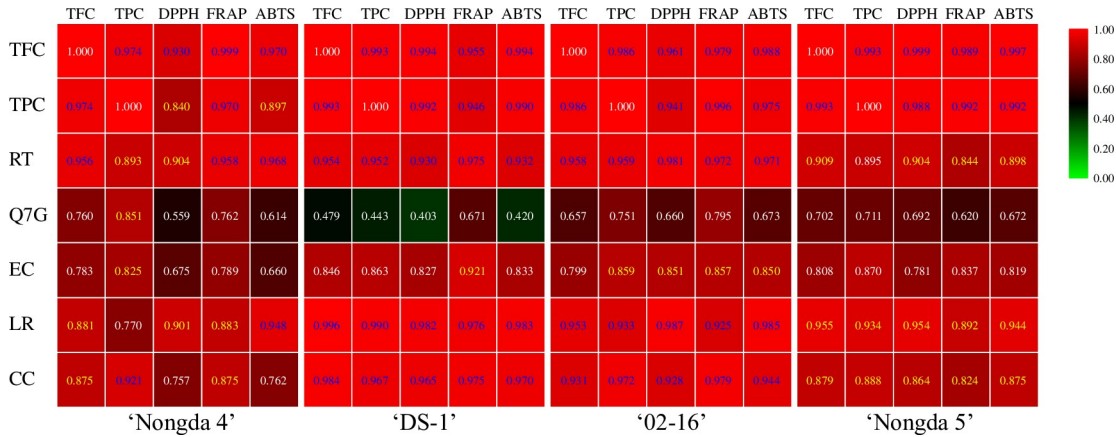

**Fig 6. Correlations among bioactive components and antioxidant indices in *Prunus humilis* fruit.** Note: Positive number, positively correlated. The blue numbers indicate a significant correlation at the 0.01 level, the yellow numbers indicate a significant correlation at the 0.05 level. TFC, total flavonoid content; TPC, total phenol content; DPPH, 2,2-diphenyl-1-picrylhydrazyl free radical scavenging capacity; FRAP, ferric reducing antioxidant power; ABTS, 2,2'-azinobis (3-ethylbenzothiazoline-6-sulfonic acid) free radical scavenging capacity; CC, catechin; EC, epicatechin; LR, liquiritigenin; RT, rutin; Q7G, quercetin-7-O-β-D-glucopyranoside.

different (Fig 6). The total phenol contents (TPC) of 'DS-1', '02–16', and 'Nongda 5' fruits were significantly positively correlated with the three antioxidant indices (*P*<0.01). The TPC of 'Nongda 4' fruit was significantly positively correlated with FRAP (*P*<0.01), and positively correlated with DPPH and ABTS (*P*<0.05). The TFC and TPC of the four *P. humilis* fruits were significantly positively correlated (*P*<0.01).

Rutin was significantly positively correlated with TFC, FRAP, and ABTS (*P*<0.01), and positively correlated with TPC and DPPH (*P*<0.05); the results show that changes in rutin content in 'Nongda 4' fruit were consistent with the changes in bioactive compound content and antioxidant capacity.

Rutin, CC and LR in the fruits of 'DS-1' and '02–16' accessions were significantly positively correlated with TFC, TPC, and antioxidant indices (*P*<0.01), and these three components showed a consistent trend with bioactive compound content and antioxidant capacity. There was no significant correlation between Q7G and TFC, TPC, or antioxidant indices.

Rutin, CC and LR were positively correlated with TFC and antioxidant indices (*P*<0.05), indicating that these three components showed the same trend as TFC and antioxidant capacity of 'Nongda 5' fruit. There was no significant correlation between Q7G and EC with TFC, TPC, or antioxidant indices.

In conclusion, the phenotype, TFC, and TPC of the four *P. humilis* fruits were different, and there were differences in the composition and content of bioactive components. These components had different effects on different antioxidant indices, so the bioactive compound content showed different trends with the antioxidant indices in different *P. humilis* fruits. In general, catechin and rutin are the most important components of *P. humilis* fruit, which is consistent with the observed changes in antioxidant capacity.

## Principal component analysis

PCA results identified two components that explained 82.09% of the total variation in bioactive components and antioxidant potential during the fruit development stage among the four *P. humilis* accessions (Fig 7). The first principal component (Prin1) contributed to 72% of the total variation. The second principal component (Prin2) contributed to 9.45% of the total variation.

Scatterplots from the PCA showed that three circles were drawn (Fig 7), and the first three development stages of 'DS-1' were brought together separately because the second principal component was lower. The first four stages of '02–16', the first three stages of 'Nongda 4', and the first two stages of 'Nongda 5' were classified into the same circle, owing to the higher components of Prin 1 and 2. With the development of fruit, the bioactive components and antioxidant potential decreased gradually, and the last two development stages of the four accessions were classified into one circle. These data suggest that from the expension stage to the turning stage, the content of total flavonoids, total phenols, individual compounds, and antioxidant indices showed a tipping point of decline.

## Discussion

Flavonoids are widely found in plants and perform various physiological functions. They are natural antioxidants with free-radical scavenging activities. In this study, during the development of *P. humilis* fruit, the flavonoid and total phenol contents in different accessions showed different trends: TFC and TPC in 'Nongda 5' gradually decreased, whereas TFC and TPC in 'Nongda 4', '02–16', and 'DS-1' increased slightly from the young fruit stage to the stone hardening stage, and then decreased. The antioxidant indices (ABTS, FRAP, and DPPH) of *P. humilis* fruit showed the same trend as those of TFC and TPC. Correlation analysis results showed that the antioxidant indices were significantly positively correlated with TFC ($P<0.01$)

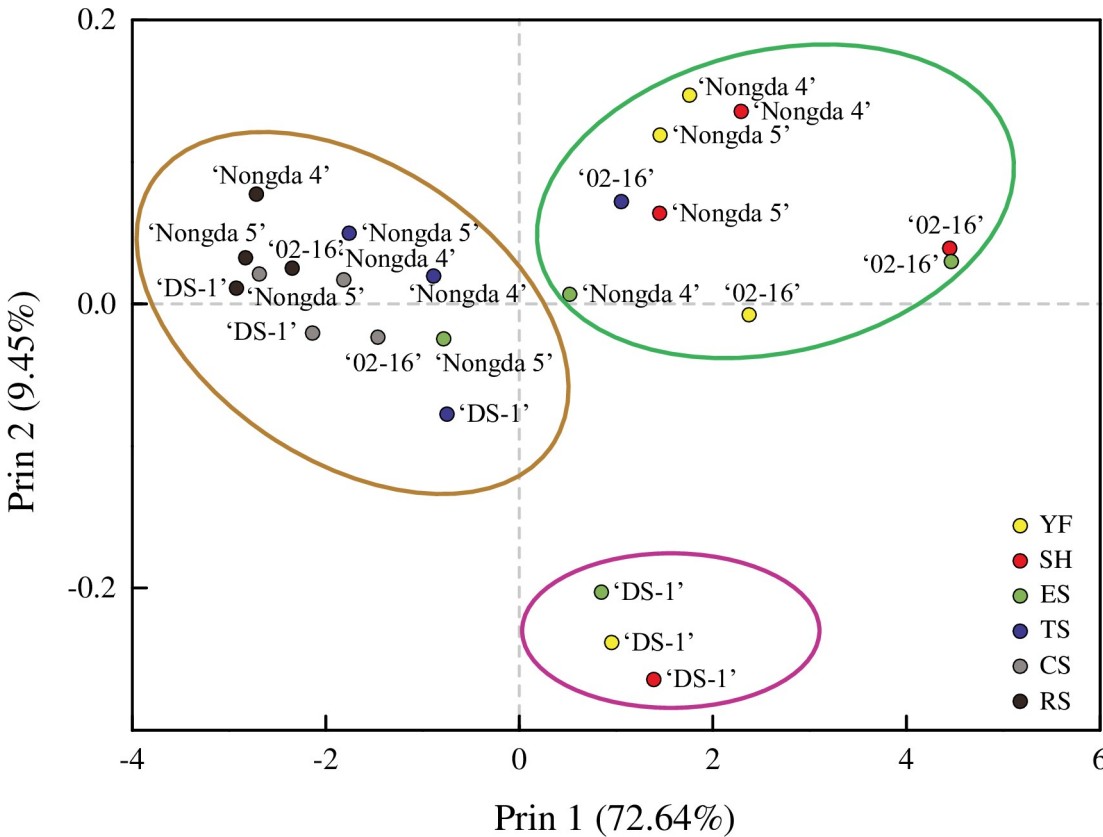

**Fig 7. Scatterplot of principal component analysis based on bioactive components and antioxidant potential during fruit development of *Prunus humilis*.** Note: YF, Young fruit stage; SH, Stone hardening stage; ES, Expension stage; TS, Turning stage; CS, Coloring stage; RS, Ripening stage. The three circles indicate the development stages belonging to the top two principal components.

and TPC ($P<0.05$), indicating that flavonoid was the main antioxidant component in *P. humilis* fruit, and that, it plays an important role in antioxidant potential. In our early research, the metabonomics of 'Nongda 4' and 'Nongda 5' were determined, and 171 components of flavonoids were detected at the ripening stage. After absolute quantitative analysis by UHPLC, CC was detected at all fruit development stages in the four *P. humilis* accessions, the CC content was ten or even one hundred times higher than that of other components, and the highest proportion of flavonoids was about 10%. It was speculated that CC may be an important flavonoid component in *P. humilis* fruit.

During the fruit development of the three pear cultivars, the content of total phenol, flavonoid, and DPPH free radical scavenging ability gradually decreased, and the total flavonoid and phenol contents were significantly positively correlated with antioxidant activity [10]. A study on the dynamic changes in flavonoid and antioxidant activity of three Kiwifruit cultivars showed that the flavonoid content decreased gradually during fruit development, and the total flavonoid content was positively correlated with the antioxidant index [11]. In a study on *Citrus grandis* (L.) Osbeck var. *tomentosa* Hort, the flavonoid content decreased sharply from 20 to 41 d, decreased slowly after 48 d, and reached the lowest value at 83 d. All development stages showed gradual flavonoid decrease [12]. Results from a different study showed that flavonoid content decreased gradually during fruit development of six Chinese jujube cultivars [13]. Flavonoid content was positively correlated with DPPH free radical scavenging ability in the study of different cultivars of Chinese jujube [13] and apricot [14]. The results of this study are consistent with those of previous studies, indicating that flavonoids accumulate in the early stage of fruit development, and that flavonoid content in the young fruit stage is abundant, but decreases during the ripening stage, which may explain why flavonoid content decreases during fruit development.

It was found that the flavonoid content in leaves and pulp had a negative correlation with fruit development in peach [15]. When a large number of flavonoids were synthesized in fruit, the precursors of flavonoids were transported into fruit in large quantity, and the flavonoid content in leaves was low; when the demand of fruit for precursors decreased, the synthesis of total flavonoids in leaves was increased. In a study of *Cerasus humilis* [16], the flavonoid content in fruit decreased gradually, while that in leaves increased gradually. The flavonoid content in 'Nongda 4' leaf was 25mg/g DW at the young fruit stage, and increased to 101.71 mg/g DW at the ripening stage, with an increase of 306.84%. In the present study, the flavonoid content in 'Nongda 4' fruit was 58.57 mg/g FW at the young fruit stage, and decreased to 11.65 mg/g FW at the ripening stage, a decrease of 402.75%, and there was a significant negative correlation between levels in leaf and fruit ($P<0.05$). Flavonoids were translocated in fruits and leaves at nutrient distribution centers, different researchers believe that there may be a transfer mechanism of flavonoid content in fruits and leaves [17, 18], which may be another reason that why flavonoid content decreases during fruit development.

There were differences in flavonoid content during the fruit development stages among the four accessions determined in this study, CC, rutin, and Q7G were detected in all fruit development stages of the four *P. humilis* fruits. Myricetin and Quercetin were detected in the early developmental stages of 'Nongda 4', 'Nongda 5', and '02–16', but were not detected in the late developmental stage. C3G was detected in the late developmental stages of 'Nongda 4' and 'DS-1', but not detected in any of the developmental stages of 'Nongda 5' and '02–16'. Correlation analysis showed that there were some differences in the correlation between flavonoid components and total flavonoid content with antioxidant indices among different accessions. Rutin, CC and LR were correlated with flavonoid content and antioxidant indices. Q7G and EC were not significantly correlated with flavonoid content or antioxidant indices. CC, EC, and RT were the main flavonoid components in the fruits of six Chinese jujube cultivars, and

they were positively correlated with flavonoid content and DPPH free radical scavenging activity; these three components are the most important antioxidants in Chinese jujube fruit [13]. During fruit development, the flavonoid content in the peel and pulp of Nanfeng tangerine decreased gradually; among the 11 flavonoid components, eight were detected in the peel and four in the pulp, which was not consistent with the observed changes in flavonoid content [19]. There are differences in flavonoid components among different species, but there is obvious correlation between flavonoid content and antioxidant indices.

Anthocyanins are a class of pigments found widely in plants, belong to the flavonoid class of compounds [20], and give plants a variety of colors. Cyanidin-3-O-glucoside is an important component of anthocyanin, and is also an important component of red color accumulation. In the present study, *P. humilis* fruit was always green in the early stage, and C3G was not detected in the first four development stages of the four accessions. At coloring stage C3G was detected in the two red accessions ('Nongda 4' and 'DS-1'), suggesting that there might be a relationship between C3G and the red color formation of *P. humilis* fruit. From 200 to 243 days after flowering, the pulp of blood orange does not appear red, and anthocyanin cannot be detected. The pulp of blood orange becomes more red and the anthocyanin content increases continuously from 261 to 324 days after flowering [21]. In mulberry, anthocyanin was the main reason for red color [22, 23], cyanidin content is the main component of mulberry anthocyanin [24]. The higher the anthocyanin content in different cultivars of *Camellia japonica*, the deeper the red color of the petal. C3G was the main anthocyanin that determined the color of *C. japonica*. Accumulation of C3G increases the degree of petal redness [25, 26]. The relative content of C3G in red sweet potato was higher than in non-red sweet potato [27]. C3G accounts for more than 63% of the total anthocyanin content in *P. humilis* fruit [7]. In conclusion, there was a close relationship between C3G and the accumulation of red color.

## Conclusions

The flavonoid content, total phenol content, and antioxidant capacity of *P. humilis* decreased gradually with fruit development. Changes in catechin, epicatechin, liquiritigenin and rutin were consistent with those of flavonoids overall. Quercetin-7-O-β-D-glucopyranoside remained stable during fruit development. Myricetin and quercetin accumulated at the early stage of fruit development, and cyanidin-3-O-glucoside began to accumulate at the coloring stage in the red accessions. In conclusion, it is better to choose a red *P. humilis* fruit for extracting cyanidin-3-O-glucoside at the ripening stage, and it is better to extract flavonoids and other components at the stone hardening stage.

## Supporting information

**S1 File.**
(XLS)

## Acknowledgments

We would like to thank Editage (www.editage.com) for English language editing. We are very grateful to the editors and reviewers for their helpful comments regarding this manuscript.

## Author Contributions

**Data curation:** Hongbo Fu, Yujia Qiao.

**Formal analysis:** Hongbo Fu.

**Funding acquisition:** Junjie Du.

**Investigation:** Hongbo Fu, Pengfei Wang.

**Methodology:** Xiaopeng Mu, Baochun Fu.

**Resources:** Pengfei Wang, Junjie Du.

**Writing – original draft:** Hongbo Fu.

**Writing – review & editing:** Jiancheng Zhang, Junjie Du.

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
