## [Decision Letter · Decision Letter 0]

12 Apr 2021

PONE-D-21-06870

Dynamic changes of bioactive components and antioxidant potential of Prunus humilis Bunge fruit

PLOS ONE

Dear Dr. Fu,

Thank you for submitting your manuscript to PLOS ONE. After careful consideration, we feel that it has merit but does not fully meet PLOS ONE’s publication criteria as it currently stands. Therefore, we invite you to submit a revised version of the manuscript that addresses the points raised during the review process.

The manuscript needs considerable restructuring, according to the Reviewers' reports, in order to headline the importance of the study. English language usage should be substantially polished. Figure captions should be completely rewritten in order to be self-explanatory. Statistical significance must be introduced in Figures 1-5. Please put labels of statistical significance in Table 2 into superscript. Do not use the entire Latin name "*Prunus humilis*" throughout the text, but instead "*P. humilis*". However, "*Prunus humilis*" should stand in the first mention in the main title, Abstract, and Introduction section. It also applies to other species' Latin names in the text, such as "*Camellia japonica*". Please precise what Nongda 4, Nongda 5, 02-16 and DS-1 represent. If they are cultivars, then they should stand under single quotation marks as 'Nongda4' and so on.

We look forward to receiving your revised manuscript.

Kind regards,

Branislav T. Šiler, Ph.D.

Academic Editor

PLOS ONE

Journal Requirements:

Reviewers' comments:

Reviewer's Responses to Questions

**Comments to the Author**

1. Is the manuscript technically sound, and do the data support the conclusions?

Reviewer #1: Yes

Reviewer #2: Yes

2. Has the statistical analysis been performed appropriately and rigorously? 

Reviewer #1: No

Reviewer #2: No

3. Have the authors made all data underlying the findings in their manuscript fully available?

Reviewer #1: Yes

Reviewer #2: Yes

4. Is the manuscript presented in an intelligible fashion and written in standard English?

Reviewer #1: Yes

Reviewer #2: No

5. Review Comments to the Author

Reviewer #1: Authors develop their work adequately on the changes of phenolics and antioxidant activities of P. humilis fruit during the development.

1. The title needs edits to reflect "the changes are during fruit development".

2. Taking into account the amount of results obtained and the different varieties used, perhaps, the article should be structured in a different way. For example, a Principal component analysis (PCA) could be conducted to see the overall differences in phenolic profile and antioxidant activities during different development stages, according to the variety studied (i.g., TPC, TFC, DPPH, FRAP, ABTS, and individual compounds)

3. Spell out the abbreviation FW for the first time, and note all the abbreviations in the legend for Figure 1-5.

4. Table 1 indicate the year.

5. Line 202 “…showed a decrease in…”

6. It looks like no significant analysis used for illustrating the results of Figure 1-5, and only mean values were compared.

7. Line 238-239 Rephrase this sentence.

Reviewer #2: The manuscript presented by Fu et al. analysed the changes in total flavonoid and anthocyanin content, the content of eight flavonoid compounds and antioxidant capacity by three methods in for accessions of Prunus humilis Bunge fruit at six develop stages. Although the manuscript presented some interesting results, it must be thoroughly reviewed before it can be accepted. Authors should place more emphasis on pointing out the importance of this study.

I advise the authors to find a native English speaker to proofread the manuscript.

The abstract should be rewritten. A brief description of the work with the most relevant results should be given. Thus, readers can get an idea of the content of the work.

Introduction. The sentences in Lines 82-84 and 84-87 are redundant.

Material and Methods.

- I suggest separating plant material and chemicals into different sections.

- Authors should include how they expressed the different results. In reference to the replicates. Please change lines 104-105 to "A total of three biological replicated consisting in 3 healthy plants each one was selected” or something like that.

- Authors should include how they expressed TFC, TPC, antioxidant capacity determined by the the three methods.

- Flavonoid content: Please indicate at which range were performed the different calibration curves. Moreover, which software were used for the acquisition data?

Results.

This section should be rewritten. Moreover, the Figure legends should be improved because Authors did not include any reference to the results showed in each one (biological samples, statiscal method, etc). They only included the titles. Furthermore, the statistical results should be included in the Figures. Based on the results showed it is difficult to know whether some of the changes are statistically different.

- Lines 289-291. Please delete. Authors should only refer to the results of this section. Such sentences should be in the discussion or in the results section where correlations are discussed.

- Lines 315-321. Please move to discussion section.

- In reference to the results included in the correlation analysis, Authors should avoid repeating that "the level of correlation is different". What is important is the biological sense of that correlation, not whether it was 0.99 or 0.93.

- Lines 337-339. Please delete.

6. PLOS authors have the option to publish the peer review history of their article (what does this mean?). If published, this will include your full peer review and any attached files.

Reviewer #1: No

Reviewer #2: No

---

## [Author Response · Author response to Decision Letter 0]

23 Apr 2021

Thank you for editor and reviewers comments, respond to reviewers has been uploaded.

---

## [Editor Report · Decision Letter 1]

26 Apr 2021

Changes of bioactive components and antioxidant potential during fruit development of Prunus humilis Bunge

PONE-D-21-06870R1

Dear Dr. Fu,

We’re pleased to inform you that your manuscript has been judged scientifically suitable for publication and will be formally accepted for publication once it meets all outstanding technical requirements.

Kind regards,

Branislav T. Šiler, Ph.D.

Academic Editor

PLOS ONE
---

## [Editor Report · Acceptance letter]

11 May 2021

PONE-D-21-06870R1 

Changes of bioactive components and antioxidant potential during fruit development of *Prunus humilis* Bunge 

Dear Dr. Du:

I'm pleased to inform you that your manuscript has been deemed suitable for publication in PLOS ONE. Congratulations! Your manuscript is now with our production department. 

Kind regards, 

on behalf of

Dr. Branislav T. Šiler 

Academic Editor

PLOS ONE